# HEBE project: Healthy aging versus inflamm-aging: The role of physical exercise in modulating the biomarkers of age-associated and environmentally determined chronic diseases, study protocol

Francesca Bianchi[1,2°], Elia Mario Biganzoli[3,4°]*, Valentina Bollati[5,6°], Mario Clerici[7,8°]*, Daniela Lucini[9,10°], Chiara Mandò[3°], Federica Rota[5°], on behalf of the HEBE Consortium[¶]

1 Department of Biomedical Science for Health, University of Milan, Milan, Italy, 2 Laboratorio Morfologia Umana Applicata, IRCCS Policlinico San Donato, Milan, Italy, 3 Department of Biomedical and Clinical Sciences L. Sacco, Medical Statistics Unit, "Luigi Sacco" University Hospital, University of Milan, Milan, Italy, 4 Data Science Research Center, University of Milan, Milan, Italy, 5 EPIGET—Epidemiology, Epigenetics and Toxicology Lab, Department of Clinical Sciences and Community Health, University of Milan, Milan, Italy, 6 Epidemiology Unit, Fondazione IRCCS Ca' Granda Ospedale Maggiore Policlinico, Milan, Italy, 7 Department of Pathophysiology and Transplantation, University of Milan, Milan, Italy, 8 Don C. Gnocchi Foundation, Istituto di Ricovero e Cura a Carattere Scientifico (IRCCS) Foundation, Milan, Italy, 9 BIOMETRA Department, University of Milan, Milan, Italy, 10 Exercise Medicine Unit, IRCCS Istituto Auxologico Italiano, Milan, Italy

☯ These authors contributed equally to this work.
¶ Membership of the HEBE Consortium is provided in the Acknowledgments.
* elia.biganzoli@unimi.it (EMB); mario.clerici@unimi.it (MC)

**Data Availability Statement:** No datasets were generated or analysed during the current study. All

## Abstract

Inflamm-aging refers to the chronic low-grade inflammation that occurs with aging and cellular senescence, and it is linked to various diseases. Understanding the markers involved in inflammation and aging, as well as their interaction with environmental factors and bodily control mechanisms, can provide crucial tools for assessing the resilience (i.e. the ability to adapt and improve) of the human body, particularly in the presence of chronic degenerative conditions or vulnerable life stages, that place the individual and the community to which he belongs in a state of potential fragility. HEBE focuses on physical exercise, along with nutritional and lifestyle recommendations, to reduce systemic inflammation and promote healthy aging. HEBE encompasses multiple research lines (LR). In the ongoing LR1 ("proof of concept"), healthy lifestyle recommendations were provided to University of Milan employees, and changes in quality of life and well-being were assessed using a specialized questionnaire. The first 100 eligible subjects, who expressed their willingness to participate, underwent a personalized physical exercise protocol based on clinical and objective assessments. Biomedical samples were collected at baseline (T0) and follow-up (T1) to establish a shared biobank and identify non-invasive biomarkers that monitor the impact of physical exercise on individual characteristics such as cardiovascular and metabolic health. Subsequently (LR2-LR10), the proof of concept findings will be expanded to include various

relevant data from this study will be made available upon study completion.

**Funding:** HEBE received support from the University of Milan (GSA, Grandi Sfide di Ateneo, Linea 6 del Piano di Sostegno alla Ricerca 2021, Award number PSRL621MCLER_01). The funders had no role in study design, data collection and analysis, decision to publish, or preparation of the manuscript.

**Competing interests:** The authors have declared that no competing interests exist.

conditions of vulnerability such as obesity, cancer, endocrine disorders, cardiovascular diseases, infertility, functional syndromes, respiratory disorders, neurodegenerative diseases, and autoimmune conditions. The research lines will leverage the expertise of the 94 participating investigators to form a collaborative network that maximizes the potential for investigation and knowledge exchange. This approach fosters a culture of health promotion and disease prevention.

ClinicalTrials.gov Identifier: NCT05815732.

## Background

Non-communicable diseases (NCDs) are the cause of premature aging and death of 41 million people each year, equivalent to 74% of all deaths globally [1], and specifically for 88% in the USA and 91% in Italy [2]. In this context, unsustainable lifestyles, often characterized by a lack of physical activity together with unsuited dietary habits, might trigger the development and progression of NCD [3]. To date, it is estimated that about 41% of Europeans (and as many as 60% of Italians) do not carry out any type of physical activity (even mild) and this increases the risk of chronic diseases [4].

Biological cellular aging has been recognized as a key factor in NCD [5]. Over the years, the progressive increase in the number of senescent cells can lead to tissue/organ dysfunction [6, 7]. Senescent cells undergo metabolic changes that lead to increased secretion of soluble factors (e.g. cytokines, chemokines, growth factors, metalloproteases, hormones) which concur to mold the microenvironment, thus affecting the composition and phenotype of the surrounding stromal cells, and promote angiogenesis and vasculogenesis processes [8, 9]. In addition, the removal of damaged cells, via recruiting phagocytes, promotes cell differentiation and proliferation for tissue renewal, regulating extracellular matrix (ECM) composition and structure [10]. Local inflammatory processes are mirrored at the systemic level, thus the secretion of pro-inflammatory molecules and activation of immune cells contribute to the generation of a chronic low-grade pro-inflammatory state known as "inflamm-aging", associated with the onset and progression of most age-related diseases [11, 12].

Counteracting inflamm-aging could be the key to ensuring healthy aging and reducing the risk of the early development of NCD. A very promising approach to impair inflamm-aging focuses on reducing the risk factors associated with NCDs by personalized lifestyle interventions. In particular, targeted physical activity can improve health, enhance the quality of life, and reduce the risk of developing NCD, resulting in both immediate and long-term health benefits [13]. Physical activity improves specific control mechanisms, such as cardiac autonomic and metabolic controls, and reduces inflammatory parameters according to specific immune responses both at the cellular and humoral levels [14–16]. In addition, extended clinical evidence shows that cardiorespiratory fitness is a stronger indicator of cardiometabolic risk factors and risk prediction than self-reported physical activity levels [17], and a powerful predictor of cardiovascular health in youth and later in life [18].

Understanding the early disease mechanisms represents the starting point for a "Medicine" that approaches the issue of environmental health in a preventive key, helping to reduce the incidence of diseases and the resulting mortality.

The ultimate goal of the HEBE project is to translate the results obtained in a clinical-pathological context and apply them in a public health context. In particular, we aim to apply the knowledge we are acquiring to counteract the negative consequences of inflammation-aging

on health in a preventive setting. This would allow us to translate the observations obtained from basic and epidemiological studies to clinical practice, allowing us to identify the subjects most susceptible to a certain risk factor (e.g. obese, elderly, people with chronic pathologies, etc.), and to indicate effective prevention strategies.

## Objectives

The main objective of the HEBE study is the evaluation of the effects of the personalized intervention based on physical exercise on clinical parameters and circulating markers in the sampling population of employees of the University of Milan.

The primary endpoint will be the increase in maximal oxygen consumption ($VO_2$ max) following the 6-month exercise program. The main secondary endpoint will be the evaluation of the effects of physical exercise on the control of the autonomic nervous system, considering the modification of the Autonomic Nervous System Index (ANSI) as the main variable. A further secondary endpoint will be the evaluation of the effect of physical exercise on the regulation of circulating levels of biomarkers. Additional endpoints will be the evaluation of the effect of physical exercise on the following parameters:

- Body composition (Percentage of fat mass and free fat mass)

- Blood chemistry parameters (e.g. blood sugar, glucose, and lipid profile)

- Improved quality of life and perception of stress symptoms

- Improved quality/quantity of sleep

- Improved psychological well-being

Furthermore, the relationship among all the biomarkers deriving from the study will be evaluated mainly at the exploratory level.

## Methods/Design

### Design and conceptual framework

HEBE is a prospective cohort started in 2022, to evaluate the role of physical exercise in modulating the inflamm-aging by determining an impact on health and on the risk of developing NCD. The study has been registered in ClinicalTrials.gov (Identifier: NCT05815732; posted on April 18, 2023). The authors confirm that all ongoing and related trials for this intervention are registered.

The applied bottom-up approach involves the cooperation of 94 researchers in life sciences and medicine at the University of Milan, to create a synergy that holistically approaches the concept of inflamm-aging and evaluates its reduction by physical exercise, integrated by lifestyle changes.

Informed consent has been and will be obtained from all study subjects before their participation. This essential ethical principle was carefully observed throughout the recruitment and data collection phases of our study. Every participant was fully informed about the nature of the research, the procedures involved, and the potential risks and benefits. All methods employed in our study were conducted in strict accordance with the relevant guidelines and regulations. HEBE received Ethics Approval from the University of Milan Ethical Committee (approval 62/22, date 30/06/ 2022).

The overall project includes 10 Lines of Research (LR) as described:

LR1 ("proof of concept"), which is currently ongoing, proposes a lifestyle modification protocol to all University of Milan employees and will be used to evaluate the changes obtained in

terms of quality of life and psychophysical well-being using questionnaires. LR1 is focused on physiological mechanisms and thus the recruitment involves only healthy subjects (accounting for cardiometabolic risk factors). We will broaden and deepen the study of biological processes at the basis of the senescence processes of the human organism, and in particular, we will define how these mechanisms change about the systemic and tissue / organ-specific inflammatory state, with physical exercise. The choice of measures that are carried out in the proof of concept, has been focused on the assumption that all levels of biological organization need to be investigated, from molecular, cellular, tissue, and organ studies, to studies that investigate model organisms as a whole (e.g., mouse model, zebrafish).

Concepts defined in LR1 will be then applied to 9 additional LRs, declining the common background of inflamm-aging and evaluating the effect of physical exercise on additional sub-cohorts according to the following lines of research (Fig 1).

- LR2: Obesity and development of comorbidities;

- LR3: Cancer;

- LR4: Endocrine origin of fragility;

- LR5: Cardiovascular diseases;

- LR6: Human fertility;

- LR7: Psychological stress;

- LR8: Respiratory diseases;

- LR9: Neuroinflammation associated with aging and frailty;

- LR10: Autoimmune diseases.

## Subjects enrolled in LR1—Phase 1

A lifestyle modification protocol has been proposed to all University of Milan employees, by sending an email from and to the institutional accounts. The University of Milan ("Università degli Studi di Milano", UNIMI) is a founder and the only Italian member of the prestigious League of European Research Universities (LERU). UNIMI employees include 2605 teachers and 1872 technical and administrative staff. Those willing to adhere to the protocol have been asked to fill in at time zero (T0) a questionnaire that will investigate lifestyle habits (physical activity, sedentary lifestyle, nutrition, smoking, perception of stress, sleep, alcohol, etc.), and clinical individual characteristics (November 8, 2022, starting of the recruitment period, expected to end December 31, 2024). After answering the questionnaire questions, participants are asked to adhere as much as possible to specific recommendations in the following 6 months. Suggestions for a lifestyle change towards healthy habits are delivered to participants through texts published on the HEBE website (https://hebe.unimi.it), which has been specifically created and curated by researchers of the University of Milan with specific expertise in the topics of interest, to guarantee that all recommendations, indications, and messages delivered are based on scientific data and evidence-based medicine (EBM). Specifically, the HEBE website has been structured with a dedicated section named "live healthy", including specific information related to physical activity, nutrition, psychophysical well-being (e.g., sleep habits, stress management, stopping smoking) and simple tools for the qualitative self-measure of health basic parameters, such as Body Mass Index, waist circumference, Healthy Diet Score, etc. This information is also provided weekly in the HEBE Social Media official Instagram profile (@hebe_unimi).

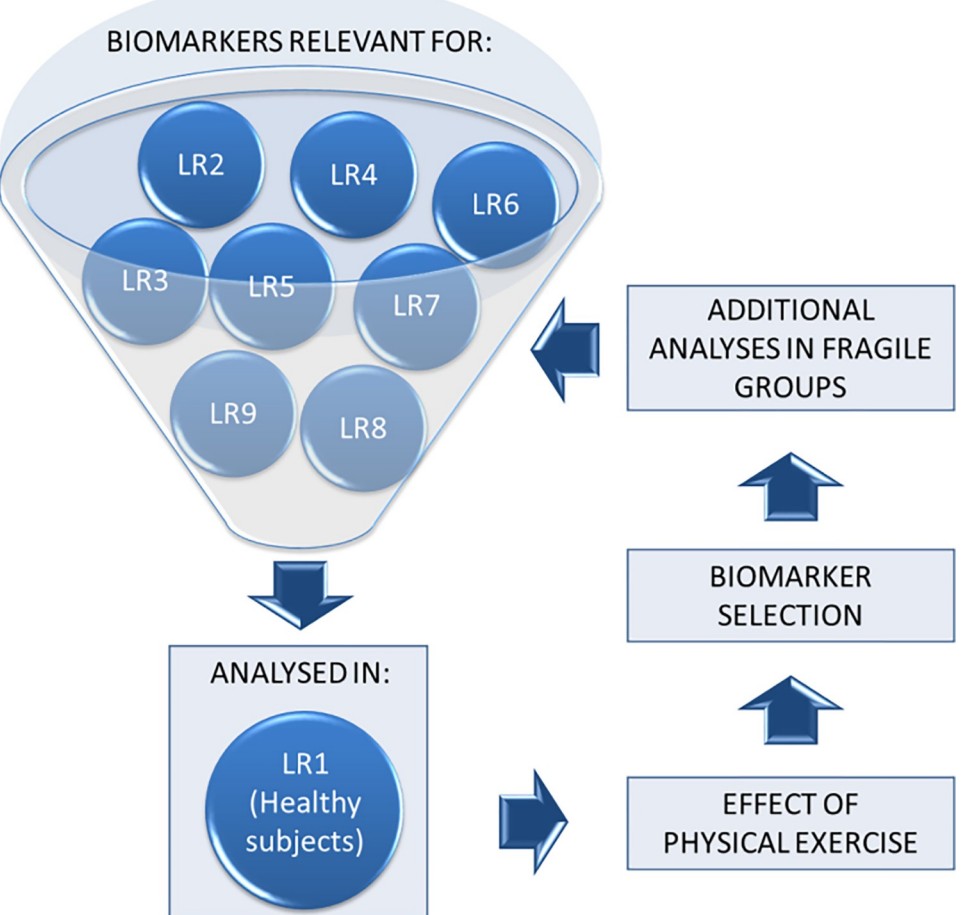

**Fig 1. Conceptual framework for the HEBE research.** The concepts established in Line of Research (LR) 1 ("proof of concept") serve as the foundation for investigating nine additional LRs. These LRs focus on distinct aspects of inflamm-aging and explore the impact of physical exercise on various sub-cohorts. The planned lines of research are as follows: LR2: Obesity and Comorbidity Development; LR3: Cancer and Inflamm-Aging; LR4: Endocrine Factors in Fragility; LR5: Cardiovascular Disease in Inflamm-Aging; LR6: Inflamm-Aging and Human Fertility; LR7: Psychological Stress and Inflamm-Aging; LR8: Respiratory Diseases and Inflamm-Aging; LR9: Neuroinflammation, Aging, and Frailty; LR10: Autoimmune Diseases and Inflamm-Aging. Each line of research extends from the foundational LR1, collectively contributing to a comprehensive understanding of inflamm-aging and its diverse implications.

After 6 months (T1), each subject is invited to fill in again the same questionnaire, integrated with additional questions to ascertain the subjects' compliance. This second assessment is aimed at investigating the changes obtained in terms of lifestyle, quality of life, and psycho-physical well-being. So far (September 20, 2023), 983 subjects filled out the questionnaire (21.9% of the involved UNIMI personnel).

At the end of this questionnaire (T0), the subjects also have the opportunity to express their willingness to be part of a personalized medical intervention (positive answer: n = 390, see Phase 2).

## Sample size

For sample size assessment [19–22] we focused on the primary objective and the endpoint related to the variation of the measures of $VO_2$ max over the observation period. Using a

Student's t-test for paired data with a level of significance $\alpha = 0.050$, a sample size of 100 pairs of pre- and post-intervention data reaches 80% power to reject the null hypothesis of no change in $VO_2$ max when the value of the effect size under alternative hypothesis is 0.28. The effect size is defined as $d = (\mu_1 - \mu_2) / \sigma$ where $\sigma$ the standard deviation of the population of pre vs post differences. According to Cohen (1988), the present effect size would be considered between low (0.2) and medium (0.5) reference values and therefore suitable for the needs of the study.

## Subjects enrolled in Phase 2

Enrolment criteria of this subgroup were: i) absence of myocardial infarction, neurodegenerative and autoimmune diseases; ii) absence of cardiac arrhythmias; iii) absence of osteo-muscular problems that prevent the execution of physical exercise; iv) not pregnant / breastfeeding; v) absence of psychiatric pathologies; vi) absence of oncological pathologies in the last 3 years treated with chemotherapy and immunotherapy). The age range of the participants who have expressed their willingness to participate to the personalized physical exercise protocol spans from 30 to 70 years. All the subjects gave their written informed consent to the study.

Among all the subjects who meet the inclusion criteria, participants will be divided according to the following grouping criteria (strata):

1. Stratification by age (50% <50 years; 50% ≥50 years);

2. Stratification by sex (50% females; 50% males);

3. Stratification by Body mass index (50% with BMI <25; 50% with BMI≥25).

Within each stratum, the final selection of the 100 volunteers is carried out based on the order of joining the study (Fig 2). These 100 volunteers are invited to a preliminary online information meeting with the investigators of the study to verify their actual intention to actively participate in all phases of the project, as described below.

A similar approach will be defined also in the next phases of the Project (beginning in 2024), as we will recruit additional subgroups of subjects with the health conditions defined in the LR2-10.

During the first visit to set up the intervention program, all subjects enrolled in Phase 2 will be asked to fill in other questionnaires relevant to the aim of the study, to collect more detailed information, concerning specific aspects not investigated by the questionnaire administered in Phase 1.

In particular:

- Mental health continuum (MHC SF) questionnaire

- The hospital anxiety and depression scale (HADS) questionnaire

- Epworth Sleepiness Scale (ESS) questionnaire

- Pittsburgh Sleep Quality Index (PSQI) questionnaire

- Fertility questionnaire (developed ad hoc for the present study).

## Biospecimen collection

The subjects who are enrolled in the personalized medical protocol are asked to donate 31 ml of blood (15 ml for biochemical analyses and 16 ml for circulating biomarker analyses), 15 ml of urine, a saliva sample, a nasal swab, and a blood card, a few days before (3–7 days) the first

## LR 1: «Proof of concept»

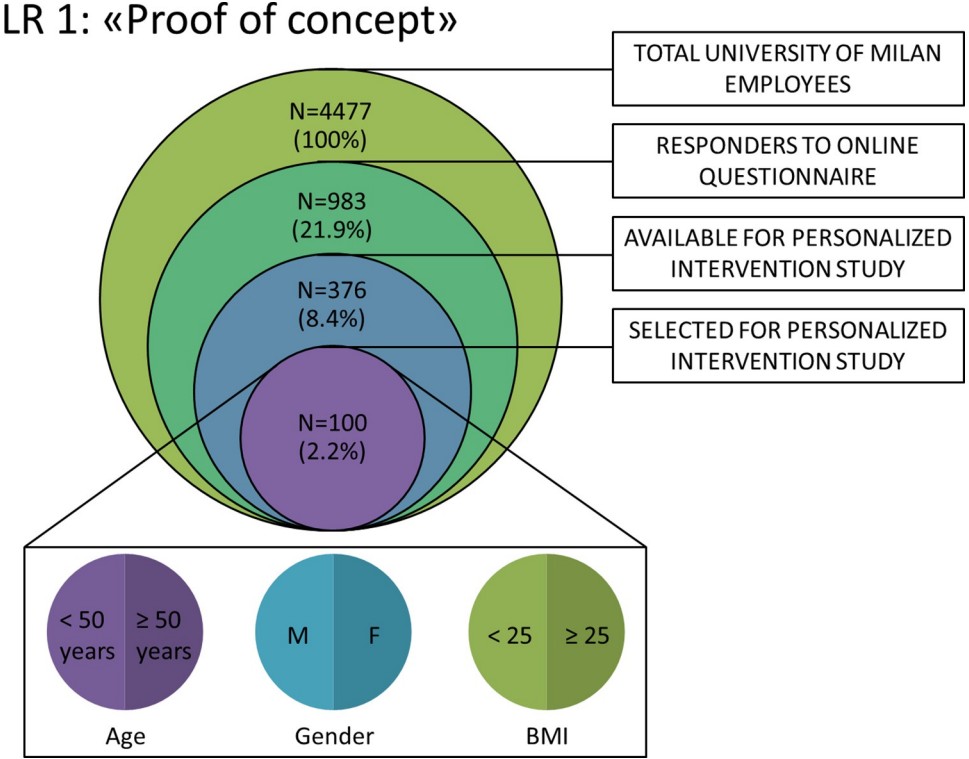

**Fig 2. Subgroup selection process for Phase 2 participants of the HEBE project.**

visit to set up the personalized protocol of physical exercise-based intervention. All samples are taken in close succession, after an overnight fasting period. Fasting blood samples are collected by a trained nurse by venipuncture in Ethylenediaminetetraacetic acid (EDTA) tubes (Becton Dickinson, Franklin Lakes, NJ, USA) between 9 and 10 a.m. and processed within 2 h from the phlebotomy. EDTA blood (16 ml) is centrifuged at 1200$g$ for 15 min at room temperature to obtain plasma. Fig 3 shows how the sample is aliquoted and stored.

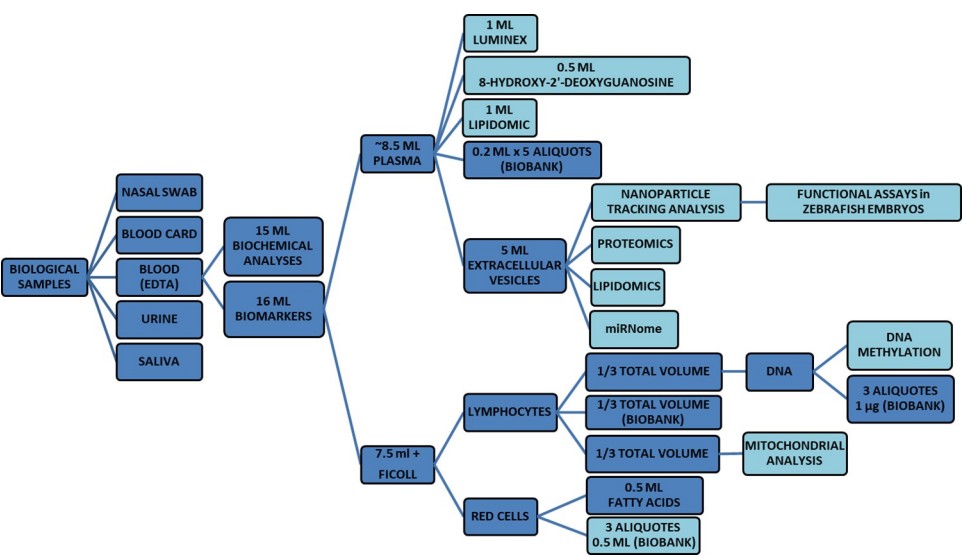

**Fig 3. Sample collection and processing workflow for participants enrolled in the personalized medical protocol.**

A 50 ml urine sample is also collected and frozen at -20˚C within 2 hours. Saliva samples are self-collected by participants using Salivette® (Sarstedt S.R.L., Nümbrecht, Germany). Briefly, the subject removes the swab from the Salivette and places the swab in the mouth, and chews it for about 60 seconds to stimulate salivation. The swab with the absorbed saliva is then returned to the Salivette and the stopper is replaced immediately. A clear saliva sample in the conical tube is obtained by centrifugation for 2 minutes at 1,000 x g yields. Nasal microbiome collection is self-performed through nasal swabs by each subject. The swabs are then immediately placed in a tube of Universal Transport Media (D.I.D. Diagnostic International, Milan, Italy) and frozen at -20˚C until DNA extraction. Finally, a drop of capillary blood is obtained from each subject by piercing the fingertip with a lancet device (Sterilance Safetylite III, VWR International S.r.l., Milan, Italy). A piece of filter paper (Whatman Protein Savercards, 903, Merck, Darmstadt, Germany) is then applied to the finger. The blood sample is dried at room temperature and then stored at -80˚C until further analyses.

## Variable collection

**Clinical assessment.**-Anthropometric parameter: Body weight is determined using a homologated electronic scale while height is measured with a statimeter, with the subject shoeless in the standing position. Body mass index (BMI) is then calculated as weight divided by the square of height ($kg/m^2$). Waist circumference is measured on the upper border of the iliac crests using a flexible graduated measuring tape.

-Hemodynamic parameters: resting systolic/diastolic Arterial Pressure and resting Heart Rate are determined using an electronic UA-611 sphygmomanometer (A&D Medical, San Jose, CA) while the subject is supine (considering at least three repeated measures) or standing up.

**Body composition.** BIA Bioelectrical Impedance Analysis (BodyStat Quadscan 4000, BodystatR_ Quadscan 4000, Body Stat Ltd., Isle of Man, British Isles) is used to estimate the percentage of fat mass (FM) and of free fat mass (FFM) using the proprietary equation provided by the manufacturer.

**Cardiac Autonomic Regulation (CAR).** Electrocardiogram (ECG), non-invasive (Finometer, TNO, Netherlands) arterial pressure, and respiratory activity (piezoelectric belt, Marazza, Italy) are acquired on a PC. Beat-by-beat data series obtained during 5 min rest and subsequently during 5 min standing are analyzed offline with dedicated software that provides time and frequency domain indices of heart rate variability (HRV) [23]. To simplify the clinical interpretation of multiple HRV variables, we proposed a unitary autonomic index (ANSI) regarded as a proxy of cardiac autonomic regulation [24].

## Lifestyle assessment

- Physical activity is assessed by a modified version of the International Physical Activity Questionnaire (IPAQ) [25, 26], which focuses on intensity (nominally estimated in Metabolic Equivalents-MET- according to the type of activity) and duration (in minutes) of physical activity. We will consider the following levels: brisk walking ($\approx$ 3.3 METs), other activities of moderate intensity ($\approx$ 4.0 METs), and activities of vigorous intensity ($\approx$ 8.0 METs) [27].

- Nutrition is assessed using the American Heart Association Healthy Diet Score [28, 29], taking into consideration fruit/vegetables, fish, sweetened beverages, whole grain, and sodium consumption.

- Sleep is assessed by inquiring about the number of hours on average slept every night and asking about the perception of sleep quality. Ad hoc questionnaires (Epworth Sleepiness Scale (ESS) questionnaire [30] and Pittsburgh Sleep Quality Index (PSQI) questionnaire [31]) are employed.

- Stress and somatic symptoms perception are assessed using a self-administered questionnaire [32, 33], providing nominal self-rated scales (higher values indicate higher degrees of symptoms) focusing on (i) the appraisal of the overall stress and fatigue perception by evaluation scales with integer scores from 0 ('no perception') to 10 ('highest perception') for each measure; (ii) the Short-Subjective Stress-related Somatic Symptoms Questionnaire (4S-Q), inquiring about four somatic symptoms accounting for the majority of somatic complaints. For scoring purposes, each response will be coded from 0 ('no feeling') to 10 ('a strong feeling'); thus, the total score ranges from 0 to 40. Moreover, perceptions of quality of life, sleep, and overall health will be assessed using evaluation scales from 0 ('worst quality') to 10 ('best quality') for each measure.

**Blood biochemical analyses.** Biochemical analyses are carried out at the IRCCS CA 'Granda Maggiore Policlinico Foundation, to assess the state of health, and the basal level of main recognized and widely used indicators of systemic inflammation, of each enrolled subject. These analyses include total cholesterol, high-density lipoprotein (HDL) cholesterol, low-density lipoprotein (LDL) cholesterol, triglycerides, Aspartate transferase (AST), alanine aminotransferase (ALT), C-reactive protein (CRP), glycemia, glycated hemoglobin, thyroid stimulating hormone (TSH), insulin, cell count, hormones (Males: testosterone, follicle-stimulating (FSH) hormone; Females: 17-beta estradiol, anti-müllerian hormone).

**Cardiopulmonary Test (CPX) and other functional tests.** Aerobic fitness for each subject is nominally expressed by the $VO_2$ max obtained with a standard incremental cycle ergometer test with a set duration of about 10±2 min, reaching a respiratory quotient $\geq 1.1$. During baseline, exercise, recovery heart rhythm, and frequency are monitored from a 12-lead ECG. Expired air is collected using a breathing apparatus and analyzed with a metabolic measurement cart to determine ventilatory and gas exchange variables on a breath-by-breath basis [34].

Moreover, we performed a 6-minute walking test to assess if the distance traveled in 6 min could be associated with the $VO_2$ max (a gold standard measure of cardiorespiratory fitness). Muscular strength is assessed using the handgrip test [35], applied both at the left and right arms.

**Circulating biomarkers.** Following the inclusive philosophy that characterizes the development of the HEBE project, during the Kick-off meeting, each LR coordinator was given the task of collecting from each of the components of the LR (for a total of 94 actively involved researchers), a list of the biomarkers of interest to be analyzed. They were also asked to support their choice through evidence derived from the literature.

The main biomarkers under investigation include measures of the DNA methylome, miR-Nome, lipidomics, proteomics, DNA oxidative damage (by ELISA), circulating proteins, mitochondrial functioning (by the Seahorse System), and telomere length (by Real-time PCR).

**Physical exercise prescription.** The prescription is performed by medical personnel, according to the clinical status of each subject, defining modalities, intensity, duration and frequency, and progression of the exercise [36]. In particular, endurance-type exercise will be prescribed to reduce cardiometabolic risk, and fat mass and increase exercise capacity. Strength-type exercise will be prescribed to improve/maintain muscle mass and flexibility-type exercise will be prescribed to improve joint flexibility and relax muscles. The dose of endurance exercise will be aimed at achieving what is recommended by the recent guidelines [37, 38]: total

Weekly physical activity volume above 600 [MET minutes/week] corresponding to at least 150 min of activity aerobic endurance per week. This dose can be exceeded as long as within 300 minutes of aerobic endurance activity per week. The protocol will be supplemented by advice aimed at stress management, smoking cessation, and proper nutrition. The subjects will then be invited to follow the personalized program at home and (for those who wish) at the Training Center of the Service Physical Exercise Medicine located at the Italian Auxological Institute, where the School of Specialization in Sports Medicine and Physical Exercise of the University of Milan is located. Contacts via web/telephone will be implemented (at least once/month), to verify what has been done, motivate the subject, and help the subject overcome barriers and difficulties. The compliance of the subjects to the execution of the physical exercise will be verified through the use of wearable devices (i.e. FitBit Inspire 3, Fitbit LLC; San Francisco, CA) that the subjects will wear for the first and last month of the intervention period. To facilitate the execution of flexibility and strength exercises, over 40 *ad-hoc* video tutorial have been recorded and delivered to each subject by the individualized prescription.

## Statistical analysis

The main outcome (variation of $VO_2$ max along the observation period) will be evaluated across the population strata and in association with the demographic, biological, and clinical covariates of interest. Descriptive analyses will be implemented to assess measured individual variations, in line with the repeated measures we collect. General linear analysis of variance/covariance models for repeated measures will then be used. Sensitivity analyses will be performed on the missing values and the robustness of the statistical models about extreme observations and assumptions on the distributions of the measures. The possibility of adopting non-parametric and/or robust regression methods of analysis will be evaluated. About the secondary/exploratory objectives regarding the biological variables considered in the project, analyses will be carried out taking into account the problem of the multiple statistical tests through the False Discovery Rate approach. Furthermore, projective techniques of multivariate analysis will be used such as Principal Component Analysis (PCA) and its non-linear and robust extensions according to inductive inference approaches (computational learning) to reduce the size of the problem and extract the most relevant information also by the structure of repeated measures of the study. A final statistical analysis plan (SAP) will be prepared during the conduct of the study before performing the final analyses.

## Dissemination of results

A project's website has been implemented and is updated regularly (https://hebe.unimi.it).

The main aim of the website is public engagement, for the promotion of a culture of health and the resulting prevention of disease, with a strong positive impact on the community.

The website is structured in sections aimed at informing citizens and patients in detail about the project and involved researchers, as well as providing recommendations related to physical activity, nutrition, and psychophysical well-being based on scientific evidence. It also strives to update the scientific most recent discoveries and Medical Society guidelines in the field of health and lifestyle and to advertise all the relevant events sponsored by HEBE.

Although the website is in Italian, a specific English section dedicated to national and international researchers is in progress, to provide useful material for scientists working in the Health field, such as the HEBE Study Protocol and scientific articles.

Moreover, information reported on the website is weekly published in the HEBE official Instagram profile (@hebe_unimi), through both posts and stories, creating a Social Community aimed at enhancing citizens' engagement and awareness.

Both the website and the social media profile will support a comprehensive approach to health literacy among the aging population, promoting the idea that lifestyle changes can positively affect individual well-being and life quality in a personalized perspective.

## Opportunities to collaborate

HEBE has been designed to be aware that much of its value would arise from involvement with other investigators, individually and within consortia. The HEBE project is open for collaboration with interested investigators. Proposals to test specific hypotheses on the HEBE cohort will be reviewed by our research team. Requests can be sent to the e-mail hebe@unimi.it.

## Data protection and management

The data are collected in a database created using the REDCap (Research Electronic Data Capture) platform Harvard Catalyst, Boston, USA. REDCap is a secure non-profit web application, protected by personal login, and allows data traceability and the implementation of databases that comply with the current General Data Protection Regulation (GDPR) legislation. Data collection takes place by the documentation relating to the RDM UNIMI research data management policy.

Moreover, even if not legally necessary, the participating members of the project have signed a consortium agreement to define and regulate the management and use of all materials/data/results collected and produced by HEBE.

## Discussion

The results produced by the HEBE Study will have a significant impact on translational and clinical research of NCDs from multiple points of view. The results produced will contribute to developing precision models for the study of the frailties underlying the main NCDs and their relationship with a sedentary lifestyle, thanks to the integration of highly multidisciplinary team members, with experimental, clinical, and computational skills and the sharing of complementary competencies. HEBE will help identify biomarkers, modulated by an integrated exercise-based intervention, and thanks to the specific clinical and preclinical skills on the frailties examined, the relationship between relevant biomarkers for each frailty and the possible modulation following the personalized intervention based on physical exercise will be defined. Moreover, it will be of interest to define the possible associations among different control mechanisms implied in the aging process and in the etiopathogenesis of NCDs.

This approach will have the highest chance of developing a rapid integration into clinical practice and transfer to the territory through institutions, companies, and associations.

Moreover, in the short term, the results produced by the "proof of concept" will have an impact on the consolidation of the interaction network between the various experts. The clinical and preclinical researchers involved in HEBE, who deal with issues of chronic inflammation/inflamm-aging and related frailties, will be able to consolidate a network of collaborations that are not based on individual initiatives but rather finds its foundations in a perspective of collaboration between different expertises. In the medium term, clinical researchers who deal with NCDs will be able to have at their disposal a further possibility of intervention, innovative, economical, and without contraindications, to improve the prognosis in patients, through a personalized pipeline. Finally, the whole society will be able to increase awareness of the benefits associated with adopting a healthy lifestyle which ultimately also leads to a reduction in the morbidity and mortality associated with NCDs.

## Acknowledgments

We extend our deepest gratitude to the participants of the HEBE study, whose commitment and cooperation are instrumental in the successful execution of this research endeavor. We would also like to express our sincere appreciation to the research groups that collaborated in this multidisciplinary effort. Their collective expertise, dedication, and collaborative spirit are essential to the design, implementation, and interpretation of this study. The full list of Members of the HEBE Consortium includes (in alphabetical order): Ambrogi Federico, Arosio Beatrice, Aureli Massimo, Baron Giovanna, Baruscotti Mirko, Battaglia Cristina, Bellocchi Chiara, Bellosta Stefano, Bernardelli Clara, Bernardelli Guseppina, Bianchi Francesca, Biganzoli Elia, Bollati Valentina, Bonomi Marco, Bravi Francesca, Brocca Paola, Brucato Antonio, Bucchi Annalisa, Camera Marina, Caporale Nicolò, Cappelletti Graziella, Casati Lavinia, Castaldi Silvana, Castiglioni Sara, Cattaneo Maria Grazia, Cazzola Roberta, Centanni Stefano, Cetin Irene, Chiaramonte Raffaella, Chiodini Iacopo, Chiricozzi Elena, Ciana Paolo, Citro Valentina, Clerici Mario Salvatore, Cogliati Chiara Beatrice, Corbetta Sabrina Luigia, D'addio Francesca, D'amato Alfonsina, Dallanoce Clelia Mariangiola Luisa, Dei Cas Michele Vittorio, Del Favero Elena, Delle Fave Antonella, Dias Rodrigues Gabriel, Edefonti Valeria, Fenizia Claudio, Ferrari Luca, Ferraroni Monica, Fiorina Paolo, Fornasari Diego Maria Michele, Fumagalli Marta, Gagliano Nicoletta, Galliera Emanuela Rita, Geginat Jens Albrecht Ernst, Giannandrea Domenica, Ingegnoli Francesca, La Porta Caterina, La Vecchia Carlo Vitantonio Battista, Lecca Davide, Lesma Elena Anna, Limonta Patrizia, Loretelli Cristian, Lucini Daniela, Mandò Chiara, Martini Valeria, Matera Carlo, Montano Nicola, Pantoni Leonardo, Papini Nadia, Paroni Rita Clara, Perego Carla, Persani Luca, Pistocchi Anna Silvia, Platonova Natalia, Podda Gianmarco, Risé Patrizia Tiziana, Riva Paola Vanda, Rizzo Angela Maria, Rondelli Valeria Maria, Rossi Marta, Rota Federica, Ruscica Massimiliano, Savasi Valeria Maria, Scavone Mariangela, Sfondrini Lucia, Sironi Luigi, Sommariva Michele, Testa Giuseppe, Tobaldini Eleonora, Tringali Cristina Alessandra, Verduci Elvira, Vicenzi Marco, Viganò Caterina Adele, Vistoli Giulio, Vitale Giovanni. All the HEBE members are affiliated with the University of Milan, Milan, Italy. e-mail: hebe@unimi.it.

## Author Contributions

**Conceptualization:** Francesca Bianchi, Elia Mario Biganzoli, Valentina Bollati, Mario Clerici, Daniela Lucini, Chiara Mandò, Federica Rota.

**Funding acquisition:** Elia Mario Biganzoli, Mario Clerici, Daniela Lucini.

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
