## [Decision Letter · Decision Letter 0]

6 Nov 2023

PONE-D-23-30486HEBE Project: Healthy aging versus inflamm-aging: the role of physical Exercise in modulating the Biomarkers of age-associated and Environmentally determined chronic diseases, Study ProtocolPLOS ONE

Dear Dr. Clerici,

Thank you for submitting your manuscript to PLOS ONE. After careful consideration, we feel that it has merit but does not fully meet PLOS ONE’s publication criteria as it currently stands. Therefore, we invite you to submit a revised version of the manuscript that addresses the points raised during the review process.

We look forward to receiving your revised manuscript.

Kind regards,

Fares Alahdab, MD, MSc

Academic Editor

PLOS ONE

2. We note that you have selected “Clinical Trial” as your article type. PLOS ONE requires that all clinical trials are registered in an appropriate registry (the WHO list of approved registries is at https://www.who.int/clinical-trials-registry-platform/network/primary-registries" https://www.who.int/clinical-trials-registry-platform/network/primary-registries and more information on trial registration is at http://www.icmje.org/about-icmje/faqs/clinical-trials-registration/). Please state the name of the registry and the registration number (e.g. ISRCTN or ClinicalTrials.gov) in the submission data and on the title page of your manuscript. a) Please provide the complete date range for participant recruitment and follow-up in the methods section of your manuscript. b) If you have not yet registered your trial in an appropriate registry, we now require you to do so and will need confirmation of the trial registry number before we can pass your paper to the next stage of review. Please include in the Methods section of your paper your reasons for not registering this study before enrolment of participants started. Please confirm that all related trials are registered by stating: “The authors confirm that all ongoing and related trials for this drug/intervention are registered”. Please see http://journals.plos.org/plosone/s/submission-guidelines#loc-clinical-trials for our policies on clinical trials. 

 [HEBE received support from the University of Milan (GSA, Grandi Sfide di Ateneo, Linea 6 del Piano di Sostegno alla Ricerca 2021)].  

5. One of the noted authors is a consortium [HEBE consortium]. In addition to naming the author group, please list the individual authors and affiliations within this group in the acknowledgments section of your manuscript. Please also indicate clearly a lead author for this group along with a contact email address.

Reviewers' comments:

Reviewer's Responses to Questions

**Comments to the Author**

1. Does the manuscript provide a valid rationale for the proposed study, with clearly identified and justified research questions?

Reviewer #1: Yes

Reviewer #2: Yes

Reviewer #3: Yes

2. Is the protocol technically sound and planned in a manner that will lead to a meaningful outcome and allow testing the stated hypotheses?

Reviewer #1: Yes

Reviewer #2: Yes

Reviewer #3: Yes

3. Is the methodology feasible and described in sufficient detail to allow the work to be replicable?

Reviewer #1: Yes

Reviewer #2: Yes

Reviewer #3: Yes

4. Have the authors described where all data underlying the findings will be made available when the study is complete?

Reviewer #1: Yes

Reviewer #2: Yes

Reviewer #3: Yes

5. Is the manuscript presented in an intelligible fashion and written in standard English?

Reviewer #1: Yes

Reviewer #2: Yes

Reviewer #3: Yes

6. Review Comments to the Author

You may also provide optional suggestions and comments to authors that they might find helpful in planning their study.

Reviewer #1: Dear Authors,

I found your study protocol for the HEBE Project: Healthy aging versus inflamm-aging: the role of physical Exercise in modulating the Biomarkers of age-associated and Environmentally determined chronic diseases very interesting.

The objectives are clearly stated and the methodology is appropriately described.

As a minor note, please double-check the manuscript for typos, grammar and spelling, as several errors were evident.

Reviewer #2: I have read with great interest this study protocol regarding the role of physical exercise, nutrition and lifestyle recommendations aimed to reduce inflammation and promote healthy aging. The main objective is very well defined and of relevant importance in counteracting non-communicable diseases. The methods section and statistical analysis are perfectly described. The whole project will be of great impact not only on the research side but even on public health awareness.

For all these reasons, the herein presented study protocol should be suitable for publication.

Reviewer #3: Plese provide comments in line 339-340. How do the authors evaluate the mitochondria fucntion, the DNA oxidative damage and the telomere length

7. PLOS authors have the option to publish the peer review history of their article (what does this mean?). If published, this will include your full peer review and any attached files.

Reviewer #1: No

Reviewer #2: No

Reviewer #3: No

---

## [Author Response · Author response to Decision Letter 0]

9 Nov 2023

We would like to express our sincere gratitude to the editors and the reviewers for the time and effort invested in reviewing our manuscript. 

We have carefully addressed all the concerns raised by the reviewers in the revised manuscript. Specifically, we have focused on improving the clarity and accuracy of the language and addressing errors.

In response to the query regarding the evaluation of mitochondria function, DNA oxidative damage, and telomere length, we have provided in the revised manuscript the name of the technique we will apply. We believe that these enhancements contribute significantly to the completeness of our study.

Furthermore, we have thoroughly revised the manuscript to adhere to the editorial guidelines of PLOS ONE, ensuring proper formatting and alignment with the journal's standards.

We appreciate the opportunity to revise and resubmit our manuscript, and we believe that the changes made have substantially improved the overall quality of the work. We hope that these modifications address the concerns raised during the review process.

We look forward to the possibility of seeing our revised manuscript accepted for publication in PLOS One.

Elia Biganzoli and Mario Clerici, on behalf of the HEBE Consortium

---

## [Decision Letter · Decision Letter 1]

2 Jan 2024

PONE-D-23-30486R1HEBE Project: Healthy aging versus inflamm-aging: the role of physical Exercise in modulating the Biomarkers of age-associated and Environmentally determined chronic diseases, Study ProtocolPLOS ONE

Dear Dr. Clerici,

Thank you for submitting your manuscript to PLOS ONE. After careful consideration, we feel that it has merit but does not fully meet PLOS ONE’s publication criteria as it currently stands. Therefore, we invite you to submit a revised version of the manuscript that addresses the points raised during the review process.

**ACADEMIC EDITOR: **Thank you. There are a few points that still require a few more details and clarifications. Please see the reviewer's comments below. We look forward to reading your revised protocol.

We look forward to receiving your revised manuscript.

Kind regards,

Fares Alahdab, MD, MSc

Academic Editor

PLOS ONE

Additional Editor Comments:

Thank you. There are a few points that still require a few more details and clarifications. Please see the reviewer's comments below. We look forward to reading your revised protocol.

Reviewers' comments:

Reviewer's Responses to Questions

**Comments to the Author**

1. Does the manuscript provide a valid rationale for the proposed study, with clearly identified and justified research questions?

Reviewer #2: Yes

Reviewer #4: Yes

2. Is the protocol technically sound and planned in a manner that will lead to a meaningful outcome and allow testing the stated hypotheses?

Reviewer #2: Yes

Reviewer #4: Partly

3. Is the methodology feasible and described in sufficient detail to allow the work to be replicable?

Reviewer #2: Yes

Reviewer #4: No

4. Have the authors described where all data underlying the findings will be made available when the study is complete?

Reviewer #2: Yes

Reviewer #4: No

5. Is the manuscript presented in an intelligible fashion and written in standard English?

Reviewer #2: Yes

Reviewer #4: Yes

6. Review Comments to the Author

You may also provide optional suggestions and comments to authors that they might find helpful in planning their study.

Reviewer #2: I congratulate with the Authors for them present study protocol who will lead to a major study of relevant importance in counteracting non-communicable diseases. The whole project will be of great impact not only on the research side but even on public health awareness one.

Reviewer #4: Bianchi et al. presented a study protocol to evaluate the effects of personalized intervention based on physical activity on clinical parameters and circulating markers in employees of the University of Milan. This ambitious project looks at various clinical parameters and circulating markers with important clinical and research implications. We congratulate the authors for the project and the interesting website. However, the methods section requires additional specificity on some topics. I have some suggestions for each part.

L139-40: In the ‘Objective’ of the study, the authors state that the intervention is based on "physical activity"; however, in the study's title, they refer to "physical exercise". Since exercise and physical activity are two different concepts, the authors should only refer to one of them.

L161: Please indicate the date of registration at ClinicalTrials.gov.

L213-14: Teachers may have different daily physical activity levels compared to workers in technical and administrative staff. According to the authors, levels of physical activity and sedentary behavior will be assessed. It would be pertinent to clarify if plans for future analyses consider analyzing the results according to the participants' levels of physical activity, as this is an important factor that could influence the results.

What are the target ages of participants?

L217 and L232: Please write the date in full.

L219: How do the authors intend to monitor the participant’s adherence to the specific recommendations presented on the website during the 6-month intervention?

L142: VO2max – confirm its spelling (VO2max or VO2max). This is also used multiple times in the manuscript and should be changed.

L236: How did the authors arrive at the value of d = 0.28 (effect size) for the sample calculation? What were the pre- and post-means and standard deviation used? We suggest that the authors specify the program used to calculate the sample size and power (if any).

L246-51: It is also known that some medications can influence the efficacy of exercise in some biomarkers such as cytokines.

L255: Is the gender correct terminology here? Or should this be sex?

L258: In this type of study, there may be a dropout rate. Therefore, recruiting an additional percentage of participants could be prudent.

L293: Please indicate the instrument that will be used to measure blood pressure.

L297: It is suggested that the authors specify the procedures to be considered when assessing body composition (for example, the time of day when the assessments will be carried out, the need to be fasting, etc...).

L277: Where and under what conditions will the biological samples be taken? Who takes the blood? It is suggested that the protocol for collecting blood, urine, saliva, and nasal swabs be described in detail. Specify whether each of these samples will be taken at the same time in the morning, whether the participants will have to be fasting (and for how long), etc. It is also suggested that the authors indicate the method and material to be used for collecting saliva, urine, and nasal swabs.

L346: Given the aim of the project, it is not clear why the authors intend to apply the 6-minute walk test in addition to the incremental cycle ergometer test.

L348: The reference of a study on the handgrip test protocol would be pertinent.

L356-8: I suggest specifying all the indicators that will be evaluated.

L360: Is exercise prescribed only by medical personnel? A multidisciplinary team including sports science professionals would be relevant.

L363-4: Please provide examples of the types of aerobic exercise that will be performed by the participants (i.e., running, walking) and how the intensity will be controlled.

L366-9: Please provide the reference that supports these recommendations.

L364: As with the recommended volume for aerobic exercise, the WHO guidelines also have specific recommendations for strength/resistance exercise. Moreover, it is important to specify the mode of strength training.

L370-73: For participants who do the exercise program at home, will they be provided with any type of material? In addition, authors should indicate if the training sessions will be supervised or unsupervised.

L376: Please indicate the model of the wearable device to be used to examine the participants' compliance with the exercise program. In addition, it would be useful to understand why participants only use the wearable in the first and last month of the intervention period.

L371: Participants will undergo a personalized training program. Will the intensity and volume be broadly the same for all participants?

Some references are not in accordance with the journal's guidelines. PLOS uses the reference style outlined by the International Committee of Medical Journal Editors (ICMJE), also referred to as the “Vancouver” style.

7. PLOS authors have the option to publish the peer review history of their article (what does this mean?). If published, this will include your full peer review and any attached files.

Reviewer #2: No

Reviewer #4: **Yes: **Pedro Duarte-Mendes

---

## [Author Response · Author response to Decision Letter 1]

10 Jan 2024

We would like to express our sincere gratitude to the editors and the reviewers for the time and effort invested in reviewing our manuscript. 

We have carefully addressed all the concerns raised by the reviewers in the revised manuscript. 

We look forward to the possibility of seeing our revised manuscript accepted for publication in PLOS One.

Elia Biganzoli and Mario Clerici, on behalf of the HEBE Consortium

Point-to point response

Reviewer #2: I congratulate with the Authors for them present study protocol who will lead to a major study of relevant importance in counteracting non-communicable diseases. The whole project will be of great impact not only on the research side but even on public health awareness one.

We thank you for your kind words and congratulations on our study protocol. We are pleased to receive your positive feedback and appreciate your recognition of the potential impact our research may have on counteracting non-communicable diseases. Your acknowledgment of the significance of our project, not only in advancing research but also in raising public health awareness, is motivating. We are committed to conducting a comprehensive and impactful study, and your encouragement reinforces our dedication to making a meaningful contribution to the field.

Reviewer #4: Bianchi et al. presented a study protocol to evaluate the effects of personalized intervention based on physical activity on clinical parameters and circulating markers in employees of the University of Milan. This ambitious project looks at various clinical parameters and circulating markers with important clinical and research implications. We congratulate the authors for the project and the interesting website. However, the methods section requires additional specificity on some topics. I have some suggestions for each part.

L139-40: In the ‘Objective’ of the study, the authors state that the intervention is based on "physical activity"; however, in the study's title, they refer to "physical exercise". Since exercise and physical activity are two different concepts, the authors should only refer to one of them.

Thank you for your observation regarding the terminology used in our study. We agree with your suggestion to use a consistent term throughout the study. After reviewing our study objectives and title, we concur that "physical exercise" more accurately reflects the nature of our intervention and amended the text accordingly.

L161: Please indicate the date of registration at ClinicalTrials.gov.

We confirm that our study is registered on ClinicalTrials.gov, and we updated our manuscript to include the specific registration date (which is April 18, 2023).

L213-14: Teachers may have different daily physical activity levels compared to workers in technical and administrative staff. According to the authors, levels of physical activity and sedentary behavior will be assessed. It would be pertinent to clarify if plans for future analyses consider analyzing the results according to the participants' levels of physical activity, as this is an important factor that could influence the results.

What are the target ages of participants?

Thank you for your suggestion regarding the consideration of participants' levels of physical activity in our analyses. We fully agree with your recommendation and recognize the potential significance of incorporating it in the model. This approach could indeed yield valuable insights into the outcomes of our study.

Regarding the target age range, our study focuses on the population of subjects employed at UNIMI, where the maximum retirement age is 72. The age range of the participants who have expressed their willingness to participate to the personalized physical exercise protocol spans from 30 to 70 years. This information will be included in the revised manuscript to provide a more comprehensive understanding of our study population.

L217 and L232: Please write the date in full.

We modified the date in full.

L219: How do the authors intend to monitor the participant’s adherence to the specific recommendations presented on the website during the 6-month intervention?

This is a very important aspect of our protocol. First, we implement regular check-ins and assessments through scheduled virtual or in-person meetings to collect self-reported data on adherence and participant experiences. These check-ins provide participants with an opportunity to share feedback, pose questions, and discuss any challenges they may encounter during the intervention. Second, we collect objective measures, such as those produced by wearable devices or activity trackers, to participants. These devices are utilized to gather quantitative data on participants' physical activity levels. This additional layer of monitoring is designed to augment the precision of adherence assessments.

It's important to note that the details you've provided were initially outlined in our paper: “Contacts via web/telephone will be implemented (at least once/month), to verify what has been done, motivate the subject, and help the subject overcome barriers and difficulties. The compliance of the subjects to the execution of the physical exercise will be verified through the use of wearable devices that the subjects will wear for the first and last month of the intervention period”.

L142: VO2max – confirm its spelling (VO2max or VO2max). This is also used multiple times in the manuscript and should be changed.

We apologize for this imprecision in our previous version of the manuscript. The revised version now maintains consistency by using “VO2 max” uniformly throughout.

L236: How did the authors arrive at the value of d = 0.28 (effect size) for the sample calculation? What were the pre- and post-means and standard deviation used? We suggest that the authors specify the program used to calculate the sample size and power (if any).

Regarding the primary objective and the endpoint related to the variation in VO2 max measurements over the observation period, a sample size of 100 pairs of pre- and post-intervention data achieves 80% power to reject the null hypothesis of no change in VO2 max when the alternative hypothesis value is 0.28, and the significance level (alpha) is 0.050, using a paired Student's t-test.

The effect size is calculated as d = (μ1 - μ1) / σ, where σ is the standard deviation of the population of pre vs post differences. According to Cohen (1988), this effect size would be considered within the low (0.2) to medium (0.5) reference range and therefore suitable for the study's needs.

Referring to the current literature, Table 1 presents some assessments of observed VO2 max values from various references. 

REFERENCE Exercise Intensity Sample size Sample description S.D. or S.E. VO2 max baseline VO2 max post-exercise Length of the study

1 Moderate 13 Young Patients Mean ± S.D. 29 ± 5.6 35.6 ± 7.3 3 months

 10 Elderly patients 17.7 ± 6.5 21 ± 4.9 

2 Moderate 14 Mean ± S.D. 35.3 ± 7.9 38.7 ± 9.1 6 weeks

3 Moderate 17 Mean ± S.D. 25.7 ± 3.2 28.1 ± 3.2 3 months

4 Moderate 15 Aerobic training Mean ± S.E. 24.8 ± 1.1 26.2 ± 0.9 3 months

 16 Anaerobic training 24.8 ± 1.8 26.8 ± 0.8 

 17 Combined training 26.5 ± 1.3 28.2 ± 0.8 

5 Moderate 25 Women Mean ± S.D. 32.8 ± 4.2 36.2 ± 5 4 months

 16 Men 39.2 ± 5.2 44.4 ± 5.3 

 25 Women 32.8 ± 4.2 39.6 ± 5.5 9 months

 16 Men 39.2 ± 5.2 48.8 ± 4.3 

 25 Donne 32.8 ± 4.2 38.8 ± 6.2 12 months

 16 Men 39.2 ± 5.2 47.9 ± 3.3 

 25 Women 32.8 ± 4.2 38.9 ± 3.2 16 months

 16 Men 39.2 ± 5.2 48.5 ± 4.1 

REFERENCES considered for power calculation

1 - Polly A. Beere, MD, PhD; Stuart D. Russell, MD; Miriam C. Morey, PhD; Dalane W. Kitzman, MD; Michael B. Higginbotham, MB - Aerobic Exercise Training Can Reverse Age-Related Peripheral Circulatory Changes in Healthy Older Men - Circulation, 1999

2 - Shannan E. Gormley, David P. Swain, Renee High, Robert J. Spina, Elizabeth A. Dowling, Ushasri S. Kotipalli, and Ramya Gandrakota - Effect of Intensity of Aerobic Training on V˙ O2max - Medicine & Science in Sport & Exercise, 2008

3 - C.J. Hass, L. Garzarella, D.V. de Hoyos, D.P. Connaughton, M.L. Pollock - Concurrent improvements in cardiorespiratory and muscle fitness in response to total body recumbent stepping in humans - European Journal of Applied Physiology, 2001

4 - Suleen S. Ho, Satvinder S. Dhaliwal, Andrew P. Hills, Sebely Pal - The effect of 12 weeks of aerobic, resistance or combination exercise training on cardiovascular risk factors in the overweight and obese in a randomized trial - BioMed Central Public Health, 2012

5 - EP Kirk, DJ Jacobsen, C. Gibson, JO Hill, JE Donnelly - Time course for changes in aerobic capacity and body composition in overweight men and women in response to long term exercise: the Midwest Exercise Trial (MET) - International Journal of Obesity, 2003

The data pertain to pre- and post-intervention measurements, and no information is provided about the distribution of observed differences. It seems reasonable to consider differences ranging from 3 to 5 units of VO2 max and a standard deviation of around 5 units of VO2 max.

In line with these assessments, a sample of 100 subjects for a total of 200 observations under the (unrealistic) assumption of independence of measurements achieves 80% power to reject the null hypothesis of no difference in VO2 max when the population mean difference is 2.0 with a standard deviation of 5.0 for both groups and a significance level (alpha) of 0.050, assuming equal variance for pre- and post-measurements. The observed effect corresponds to an effect size of 0.4 greater than that of the paired t-test, not considering the increased precision due to paired measurements, a worst-case scenario. Given the nature of the paired measurements in this study, it is conceivable that the study could detect even smaller differences in VO2 max with adequate statistical power.

L246-51: It is also known that some medications can influence the efficacy of exercise in some biomarkers such as cytokines.

Thank you for bringing up this important consideration. We acknowledge that medications can potentially influence exercise efficacy and impact biomarkers, including cytokines. To address this, we would like to clarify that all medications taken by participants are systematically recorded, and we plan to thoroughly evaluate them as potential covariates for adjustment in our analyses. 

L255: Is the gender correct terminology here? Or should this be sex?

We agree with you, the right term should be sex. We amended the text accordingly.

L258: In this type of study, there may be a dropout rate. Therefore, recruiting an additional percentage of participants could be prudent.

Thank you for your insightful comment regarding the potential dropout rate in our study. We acknowledge the importance of addressing this concern and, in theory, agree that recruiting an additional percentage of participants could be prudent to account for potential attrition. However, it is essential to note that our approved protocol by the ethics committee specifies a sample size of 100 subjects. Adhering to ethical guidelines and study protocols, we are unable to recruit beyond this approved number to maintain the integrity of the study and adhere to ethical guidelines.

L293: Please indicate the instrument that will be used to measure blood pressure.

As suggested, we indicated that we use an electronic UA-611 sphygmomanometer (A&D Medical, San Jose, CA).

L297: It is suggested that the authors specify the procedures to be considered when assessing body composition (for example, the time of day when the assessments will be carried out, the need to be fasting, etc...).

Thank you for your suggestion regarding the need to specify procedures for assessing body composition. 

Participants in our study may have consumed a light meal before body composition assessments. We acknowledge that, unlike blood draws which are standardized in terms of fasting and timing, the medical visits were not standardized regarding the time of day and fasting status. This decision was made intentionally to accommodate the needs and convenience of the participating patients.

However, we are mindful of the importance of collecting information on the time of medical visits, and we recorded this information to assess potential confounding effects. During the analysis process, we will carefully examine this information to better understand the context of body composition assessments and evaluate how any variations might impact the results.

L277: Where and under what conditions will the biological samples be taken? Who takes the blood? It is suggested that the protocol for collecting blood, urine, saliva, and nasal swabs be described in detail. Specify whether each of these samples will be taken at the same time in the morning, whether the participants will have to be fasting (and for how long), etc. It is also suggested that the authors indicate the method and material to be used for collecting saliva, urine, and nasal swabs.

The collection of biological samples, including blood, urine, saliva, and nasal swabs, is a crucial aspect of our study design. We acknowledge the necessity of detailing the collection procedures. Thus we modified the protocol as follows:

“The subjects who are enrolled in the personalized medical protocol are asked to donate 31 ml of blood (15 ml for biochemical analyses and 16 ml for circulating biomarker analyses), 15 ml of urine, a saliva sample, a nasal swab, and a blood card, a few days before (3-7 days) the first visit to set up the personalized protocol of physical exercise-based intervention. All samples are taken in close succession, after an overnight fasting period. Fasting blood samples are collected by a trained nurse by venipuncture in Ethylenediaminetetraacetic acid (EDTA) tubes (Becton Dickinson, Franklin Lakes, NJ, USA) between 9 and 10 a.m. and processed within 2 h from the phlebotomy. EDTA blood (16 ml) is centrifuged at 1200g for 15 min at room temperature to obtain plasma. 

A 50 ml urine sample is also collected and frozen at -20°C within 2 hours. Saliva samples are self-collected by participants using Salivette® (Sarstedt S.R.L., Nümbrecht, Germany). Briefly, the subject removes the swab from the Salivette and places the swab in the mouth, and chews it for about 60 seconds to stimulate salivation. The swab with the absorbed saliva is then returned to the Salivette and the stopper is replaced immediately. A clear saliva sample in the conical tube is obtained by centrifugation for 2 minutes at 1,000 x g yields. Nasal microbiome collection is self-performed through nasal swabs by each subject. The swabs are then immediately placed in a tube of Universal Transport Media (D.I.D. Diagnostic International, Milan, Italy) and frozen at -20°C until DNA extraction. Finally, a drop of capillary blood is obtained from each subject by piercing the fingertip with a lancet device (Sterilance Safetylite III, VWR International S.r.l., Milan, Italy). A piece of filter paper (Whatman Protein Savercards, 903, Merck, Darmstadt, Germany) is then applied to the finger. The blood sample is dried at room temperature and then stored at -80 °C until further analyses.

L346: Given the aim of the project, it is not clear why the authors intend to apply the 6-minute walk test in addition to the incremental cycle ergometer test.

We appreciate your inquiry regarding the inclusion of both the 6-minute walk test and the incremental cycle ergometer test in our study. Our rationale for incorporating both tests stems from the desire to explore potential correlations between them. While the incremental cycle ergometer test is a comprehensive measure, its complexity may limit its feasibility in routine clinical practice. By also utilizing the 6-minute walk test, which is simpler and more practical, we aim to assess whether it exhibits correlations with the various biomarkers measured in the HEBE subjects, particularly in relation to clinical or pre-clinical markers. This dual-test approach allows us to capture a more comprehensive understanding of physical performance and potential relationships with biom

---

## [Decision Letter · Decision Letter 2]

20 Feb 2024

HEBE Project: Healthy aging versus inflamm-aging: the role of physical Exercise in modulating the Biomarkers of age-associated and Environmentally determined chronic diseases, Study Protocol

PONE-D-23-30486R2

Dear Dr. Clerici,

We’re pleased to inform you that your manuscript has been judged scientifically suitable for publication and will be formally accepted for publication once it meets all outstanding technical requirements.

Kind regards,

Fares Alahdab, MD, MSc

Academic Editor

PLOS ONE

Additional Editor Comments (optional):

Reviewers' comments:

Reviewer's Responses to Questions

**Comments to the Author**

1. Does the manuscript provide a valid rationale for the proposed study, with clearly identified and justified research questions?

Reviewer #4: Yes

2. Is the protocol technically sound and planned in a manner that will lead to a meaningful outcome and allow testing the stated hypotheses?

Reviewer #4: Yes

3. Is the methodology feasible and described in sufficient detail to allow the work to be replicable?

Reviewer #4: Yes

4. Have the authors described where all data underlying the findings will be made available when the study is complete?

Reviewer #4: Yes

5. Is the manuscript presented in an intelligible fashion and written in standard English?

Reviewer #4: Yes

6. Review Comments to the Author

You may also provide optional suggestions and comments to authors that they might find helpful in planning their study.

Reviewer #4: The authors answered all suggestions and questions. This interesting project looks at various clinical parameters and circulating markers with substantial clinical and research implications. I recommend this manuscript for publication.

7. PLOS authors have the option to publish the peer review history of their article (what does this mean?). If published, this will include your full peer review and any attached files.

Reviewer #4: **Yes: **Pedro Duarte-Mendes
